

# 1 Response of Coastal California Hydroclimate to the Paleocene-
# 2 Eocene Thermal Maximum

Xiaodong Zhang[1*], Brett J. Tipple[2], Jiang Zhu[3], William D. Rush[4], Christian A. Shields[3], Joseph
B. Novak[5], James C. Zachos[1]
[1]Department of Earth and Planetary Sciences, University of California, Santa Cruz, CA 95064, USA
[2]FloraTrace Inc., Salt Lake City, UT 84103, USA
[3]Climate and Global Dynamics Laboratory, National Center for Atmospheric Research, Boulder, CO 80307, USA
[4]Department of Environmental Studies and Sciences, Santa Clara University, Santa Clara, CA 95053, USA
[5]Department of Ocean Sciences, University of California, Santa Cruz, CA 95064, USA
*Correspondence to: Xiaodong Zhang (xzhan335@ucsc.edu)
**Abstract.** The effects of anthropogenic warming on the hydroclimate of California are becoming
more pronounced, with increased frequency of multi-year droughts and flooding. As a past
analog for the future, the Paleocene-Eocene Thermal Maximum (PETM) is a unique natural
experiment for assessing global and regional hydroclimate sensitivity to greenhouse gas
warming. Globally, extensive evidence (i.e., observations, climate models with high $p$CO$_2$)
demonstrates hydrological intensification with significant variability from region to region (i.e.,
dryer or wetter, or greater frequency and/or intensity of extreme events). Central California
(paleolatitude ~42°N), roughly at the boundary between dry subtropical highs and mid-latitude
low pressure systems, would have been particularly susceptible to shifts in atmospheric
circulation and precipitation patterns/intensity. Here, we present new observations and climate
model output on regional/local hydroclimate responses in central California during PETM. Our
findings based on multi-proxy evidence within the context of model output suggest a transition to
an overall drier climate punctuated by increased precipitation during summer months along the
central coastal California during the PETM.
**1 Introduction**



Global warming of a few degrees Celsius over the next century is projected to intensify the
hydrological cycle on a range of temporal and spatial scales, manifested primarily by amplified
wet-dry cycles(Held and Soden, 2006; Douville et al., 2021). Indeed, just over last few decades
there has been an increasing frequency in the severity of extremes characterized by compound
heat waves and intense drought(Büntgen et al., 2021; Williams et al., 2020; Zscheischler and
Lehner, 2022), and/or heavy precipitation and flooding(Liu et al., 2020; Risser and Wehner,
2017). As greenhouse gas driven warming continues, such precipitation extremes (wet or dry)
are expected to intensify(Stevenson et al., 2022).
California, a region vulnerable to amplified wet-dry cycles, is already experiencing multiyear
extreme droughts with longer precipitation deficits interspersed with anomalously wet
years(Zamora-Reyes et al., 2022). For example, the prolonged drought from 2012 to 2016
preceded the exceptionally high numbers of atmospheric river storms-related winter flooding of
2017(Simon Wang et al., 2017). Collectively, climate models (e.g., CESM, CMIP etc.) show that
the occurrence of such extremes in droughts and excessive seasonal precipitation in California is
expected to increase by the end of the century(Vogel et al., 2020; Swain et al., 2018). In addition,
such 'whiplash' hydroclimate shifts related to anthropogenic warming are generally supported by
historical records of California climate cycles(de Wet et al., 2021; Polade et al., 2017).
The most robust evidence for greenhouse warming-induced intensification of the hydrological
cycle comes from global warming events of the deep past(Carmichael et al., 2017). In particular,
the Paleocene-Eocene Thermal Maximum (PETM) has emerged as a unique natural experiment
for assessing global and regional hydroclimate sensitivity to greenhouse gas warming(Zachos et
al., 2008). Extensive evidence exists for a major mode shift of local/regional precipitation
patterns and intensity(Pagani et al., 2006; Slotnick et al., 2012; Schmitz and Pujalte, 2003; Sluijs
and Brinkhuis, 2009; Smith et al., 2007; Handley et al., 2012; Kozdon et al., 2020) including
enhanced erosion and extreme flooding in fluvial sections (e.g., Pyrenees; Bighorn basin), and
increased weathering and sediment fluxes to coastal basins (e.g., Bass River, Wilson Lake, mid-
Atlantic coast; Mead Stream, New Zealand etc.) along with other observations(John et al., 2008;
Nicolo et al., 2010; Stassen et al., 2012; Self-Trail et al., 2017; Wing et al., 2005; Kraus and
Riggins, 2007; Foreman, 2014).



These observations of regional hydroclimate change serve as the basis for climate model
experiments forced with proxy-based estimates of $\Delta p\mathrm{CO_2}$ for the PETM (i.e., 3x-6x pre-
industrial)(Kiehl and Shields, 2013; Carmichael et al., 2016; Zhu et al., 2020). Using such
estimates, model simulations show an overall increase in poleward meridional water vapor
transport as manifested by a net increase in evaporation of subtropical regions, balanced by
higher precipitation of tropical/high latitudes characterizing the 'wet-gets-wetter and dry-gets-
drier' hydrological response. The latest simulations using high-resolution climate models display
several key regional responses including increased frequency of extreme precipitation events,
especially the coastal regions where atmospheric rivers (AR) are common(Rush et al., 2021).
Indeed, observations of high-energy flooding events in SW Europe (i.e., the Pyrenees) during the
PETM(Schmitz and Pujalte, 2003) can be explained by increased frequency of North Atlantic
ARs contributing landfall in that region. Pacific AR activity as simulated for the PETM also
becomes more intense but less frequent along the central California coast by shifting northward
with the storm tracks(Shields et al., 2021), not unlike the projections for California in the
future(Shields and Kiehl, 2016; Massoud et al., 2019). This pattern is consistent with warming
scenarios in general which have weakened zonal wind belts (i.e., the westerlies) that are shifting
poleward(Abell et al., 2021; Douville et al., 2021).
Testing the theoretical response of Northeast Pacific ARs and seasonal precipitation along North
America's western coast in general is challenging and still limited by the lack of observations.
Here we constrain the regional hydroclimate response along the central California coast during
the PETM using several independent proxies (i.e., clay mineralogy, grain size distribution,
$\delta^{13}\mathrm{C_{org}}$ stratigraphy, and leaf wax $\delta^2\mathrm{H_{n\text{-}alkane}}$ isotope records), which are either directly or
indirectly sensitive to shifts in precipitation patterns/intensity. These proxies are then compared
against sophisticated Earth System model simulations of the PETM climate to characterize the
relative changes in regional precipitation (i.e., pattern/intensity). The new records complement
data from a previous study(John et al., 2008), and along with the latest climate modeling
experiments provide a unique case study of the sensitivity of regional hydroclimate to major
greenhouse warming.
**2 Materials and methods**

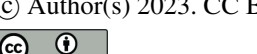



## 2.1 Site Location

The studied outcrop section is part of the late Paleocene-early Eocene Lodo Formation located in the Panoche Hill of central California (Fig. 1). During the late Paleocene, the section was situated at a paleolatitude ~42°N, roughly at the boundary between the dry subtropical highs and mid-latitude low-pressure systems. The Lodo Formation is comprised primarily of siltstone with a relatively low abundance of calcareous microfossils truncated by thin glauconitic sand layers(Brabb, 1983). Depositional facies are consistent with neritic-bathyal setting along the outer shelf(John et al., 2008).

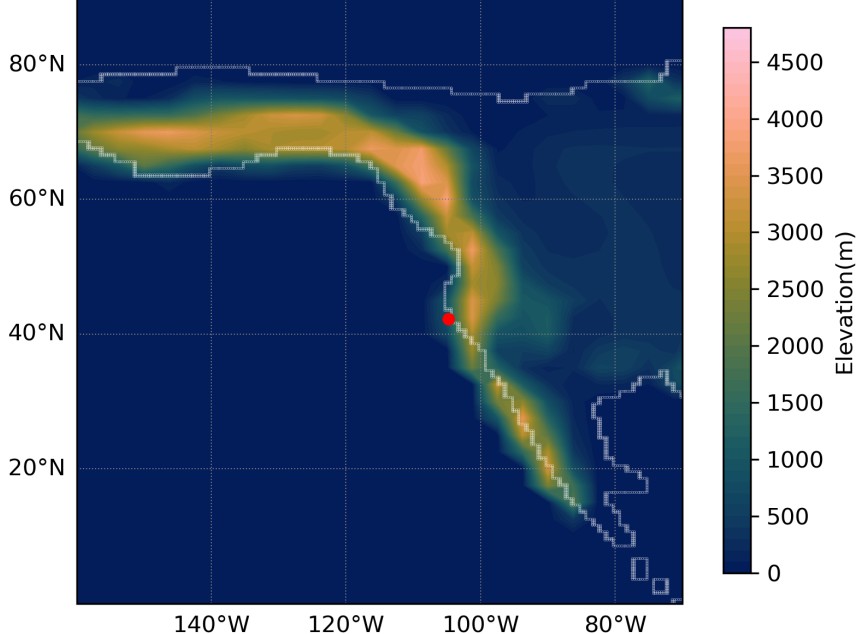

Figure 1. Paleogeography and location of the Lodo Gulch section (red spot) along the Pacific coast at 56 Ma. Late Paleocene-early Eocene topography of North America is adapted from Lunt et al. (2017).



2.2 Methods
2.2.1 Stable isotopes
Sediment samples used for this study include those originally collected by John et al. (2008).  In
addition, new samples were collected from the upper Paleocene for organic C isotopic analysis to
better establish pre-PETM baseline. Samples were analyzed in the UCSC Stable Isotope
Laboratory using a CE instruments NC2500 elemental analyzer coupled with Thermo Scientific
Delta Plus XP iRMS via a Thermo-Scientific Conflo III. All samples are calibrated with VPDB
(Vienna PeeDee Belemnite) for $\delta^{13}C$ and AIR for $\delta^{15}N$ against an in-house gelatin standard
reference material (PUGel). Analytical reproducibility precision is $\pm$ 0.1 ‰ for $\delta^{13}C$ and $\pm$ 0.2 ‰
for $\delta^{15}N$.
2.2.2 Grain Size analysis
Particle size was measured by laser diffraction using Beckman Coulter with Polarization
Intensity Differential Scatter (PIDS) housed at UCSC. 2 to 5 mg of bulk sediments was powered
in each sample for measurement. Each sample was performed 2 or 3 replicates to ensure
reproducibility.
2.2.3 Clay Assemblages analysis
Sample preparation follows a slightly modified version of (Kemp et al., 2016). Roughly 5 to 10 g
of sediment was powdered in a pestle and mortar and then placed in a Calgon (Sodium
hexametaphosphate) solution on a shaker table for 72 hours. Samples were sorted through a 63
μm sieve while collecting the fluid with the <63 μm fraction. The collected fluid and suspended
fine fraction (< 63 μm) were allowed to settle for a period determined by Stokes' Law to keep <
2um size clay particles remaining in suspension. The fluid is then decanted and dried in the oven.
Approximately 150 mg clay of each sample is used to prepared oriented mounts for X-ray
diffraction (XRD) analysis. A total of 38 clay samples were prepared from the Lodo Formation.
The sample residues are measured on a Philips 3040/60 X'pert Pro X-ray diffraction instrument
at UCSC. Clay species are identified based on peak positions and intensities representing each
clay mineral.
2.2.4 Leaf wax



Sediment extraction, compound isolation, and compound-specific isotope measurements were
conducted following Tipple et al., 2011. Briefly, sediments were freeze-dried, powdered (~500
g), and extracted with dichloromethane (DCM): methanol (2:1, v/v) using a Soxhlet extractor.
Total lipid extracts were concentrated and then separated by column chromatography using silica
gel. Normal-alkanes were further purified from cyclic and branched alkanes using urea adduction
following (Wakeham and Pease, 2004). Normal-alkane abundances were determined using gas
chromatograph (GC) with a flame ionization detector (FID). Isotope analyses were then
performed using a GC coupled to an isotope ratio mass spectrometer interfaced with a GC-C III
combustion system or a High Temperature Conversion system for $\delta^{13}C$ and $\delta^{2}H$ analyses,
respectively. $\delta^{13}C$ and $\delta^{2}H$ values are expressed relative to Vienna Pee Dee belemnite (VPDB)
and Vienna Standard Mean Ocean Water (VSMOW). Individual n-alkane isotope ratios were
corrected to n-alkane reference materials (for $\delta^{13}C$, C20, C25, C27, C30, and C38 of known
isotopic ratio and for $\delta^{2}H$, "Mix A" from Arndt Schimmelmann, Indiana University) analyzed
daily at several concentrations. In addition, $H_2$ reference gas of known isotopic composition was
pulsed between sample n-alkane peaks to confirm if normalizations were appropriate. Standard
deviations (SD) of n-alkane reference materials was $\pm 0.6‰$ for $\delta^{13}C$ and $\pm 6‰$ for $\delta^{2}H$.
2.2.5 Earth System Models
Two different set of climate simulations were used in this paper for (1) comparison with leaf wax
proxy data and (2) extreme events analysis. (1) Water isotope-enabled Community Earth System
Model version 1.2 (iCESM1.2) simulates changes in climate and water isotopic composition
during the PETM (Zhu et al., 2020) with a horizontal resolution of 1.9×2.5° in atmosphere and
land, and a nominal 1 degree in the ocean and sea ice components. Water isotope capabilities
have been incorporated into all the components of CESM 1.2 (Brady et al., 2019), which include
the Community Atmosphere Model, version 5(CAM5) for the atmosphere, the Parellel Ocean
Program, version 2(POP2) for the ocean, the Community Land Model, version 4(CLM4) for the
land, River Transport Model (RTM) for river flow, and Community Ice Code, version 4 for sea
ice. (2) Using the same CESM1.2 framework, high resolution (0.25°) simulations were
conducted with forced sea surface temperatures (SSTs) and active atmosphere and land
components (CAM5, CLM4). RTM was run at 1° resolution, and forced SST were calculated
from consistent 2° fully coupled PETM simulations (see details in Rush et al., 2021 and





reference therein). The much higher horizontal resolution in the atmosphere enables improved
simulation of the extreme events. Hourly, daily (CAM5), and monthly(iCESM1.2) temporal
resolution precipitation outputs from both sets of climate simulations were utilized in this paper,
with 100 years taken from the equilibrated iCESM1.2 simulations, and 15 years from the forced
SST high resolution CAM5 simulations.

**3 Results**
3.1 Carbon isotopes
A carbon isotope excursion is present in both bulk organic and carbonate based $\delta^{13}C$ records
across the P-E boundary (Fig. 2a), marking the PETM onset of the Lodo section(John et al.,
2008). The terrestrial leaf wax n-alkane record captures the carbon isotope excursion (CIE) with
a pattern that roughly parallels the other records. The magnitude of the $\delta^{13}C_{n\text{-alkane}}$ change is
roughly 4 ‰ (average of n-$C_{27}$, n-$C_{29}$, n-$C_{31}$) at the onset, followed by a gradual recovery that is
truncated marking the top of the PETM body (Fig. 3b).

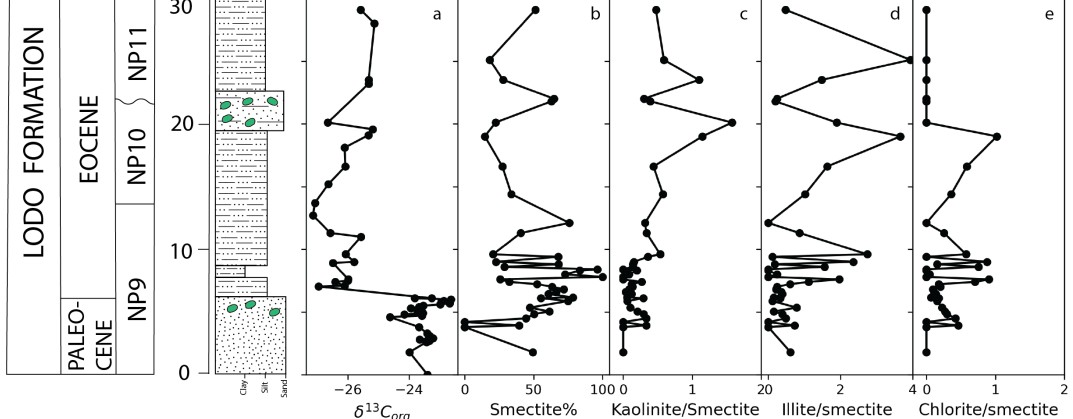

Figure 2. Integrated C isotope and clay assemblage records of Lodo Fm in the Lodo Gulch of
central California (a) bulk organic carbon isotope, (b,c,d,e) clay assemblage ratios.







3.2 Hydrogen isotopes
The leaf wax $\delta^2H_{n-alkane}$ values decrease by 25‰ just prior to the CIE onset followed by a slight
enrichment of in the main body PETM (Fig. 3c). The relatively invariable $\delta^2H_{n-alkane}$ through the
PETM is punctuated with one or two brief intervals of more negative values.

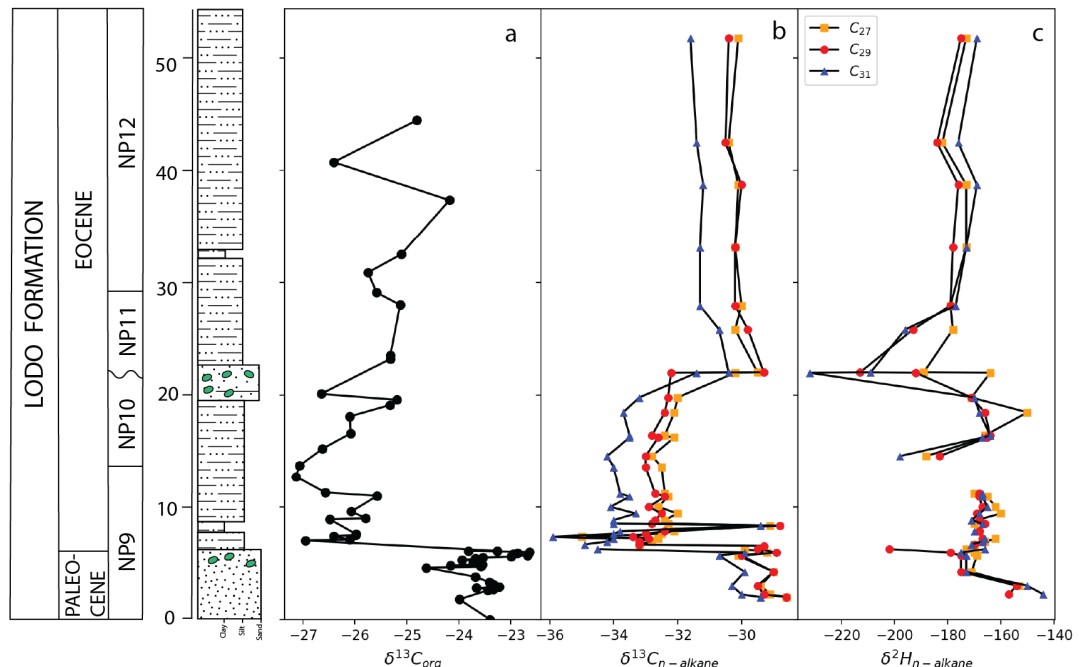


Figure 3. Marine $\delta^{13}C$ and terrestrial higher plant leaf wax n-alkane $\delta^{13}C$ and $\delta^2H$ records. (a)
bulk organic carbon isotope record of Lodo Fm. (b,c) leaf wax compound specific
carbon/hydrogen isotope records in n-C$_{27}$(yellow), n-C$_{29}$(red), n-C$_{31}$(blue).

3.3 Clay Assemblage and grain Size
Clay assemblage data (Fig. 2) shows smectite dominated in the Lodo Formation during PETM,
along with increasing illlite/smectite and chlorite/smectite ratios. The lower Lodo Formation,
with relative coarse sandy size, shows slight spikes of kaolinite associated with other minerals.
Grain size, largely silt and clay, shows a distinct shift toward finer fractions (i.e., clay) with the
onset of the CIE (Fig. S1).



3.4 Earth system model simulations

Precipitation output from two earth system models: isotope enabled iCESM1.2 under enhanced greenhouse gas simulations (1x,3x,6x,9x $p$CO$_2$ pre-industrial) and high-resolution CAM5 models (daily precipitation over 15 years), were analyzed. For this study we used 3x to 6x $p$CO$_2$ forcing that best replicate the observed ΔSST from pre-PETM to PETM(Zhu et al., 2020). Overall, monthly precipitation for the study region decreases during the PETM in both simulations but with a slight increase in the summer (Fig. 4,5). CAM5 output shows a modest decrease in mean annual precipitation with significant seasonal shifts during PETM (Fig. 5a). Seasonal changes of monthly averaged $\delta^{18}$O and $\delta^2$H from mean monthly precipitation (MAP) in iCESM1.2 of central California are consistent with CAM5. On average the $\delta^2$H increases by ca. 10 ‰ from pre-PETM to PETM, especially in the winter/spring, with a smaller shift in summer/fall (Fig 4. a,b,c). The Extreme value index (ξ), a representation of the distribution of exceedance right tail (supplemental information), shows a small but statistically robust increase in wet extremes of winter (DJF) with a significant increase in summer (JJA) wet exceedances during the PETM in the precipitation output from CAM5 simulations (Fig. 5b).

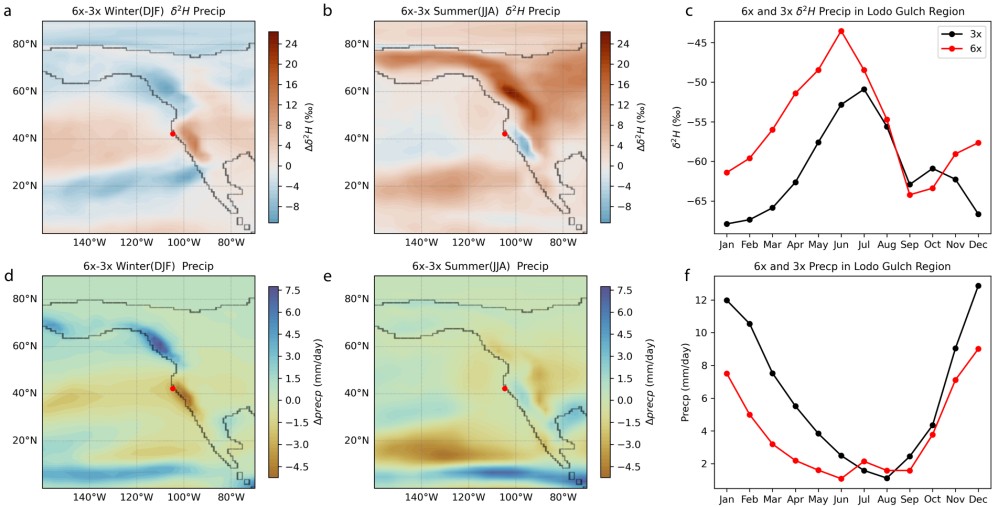

Figure 4. Water isotope-enable iCESM1.2 model output (Zhu et al., 2020) of monthly hydrogen isotope record of meteoric precipitation under different pCO$_2$ forcing to pre-industry (3x in black





represents pre-PETM, 6x in red represents PETM) in east Pacific coast (red spot represents the
Lodo Gulch study site). The iCESM1.2 simulations used the DeepMIP boundary conditions
(Lunt et al., 2017).  Difference in winter (a) and summer (b) hydrogen isotope composition
between pre-PETM(3x) and PETM(6x). (c) Annual seasonal cycle of hydrogen isotope
composition of precipitation in Lodo Gulch region. Difference in winter(d) and summer(e)
precipitation amount between pre-PETM(3x) and PETM(6x) in east Pacific coast. (f) Annual
seasonal cycle of precipitation amount in Lodo Gulch region. Site values of Lodo Gulch region
are calculated by area-weighted average over 4° x 4° box around study site.

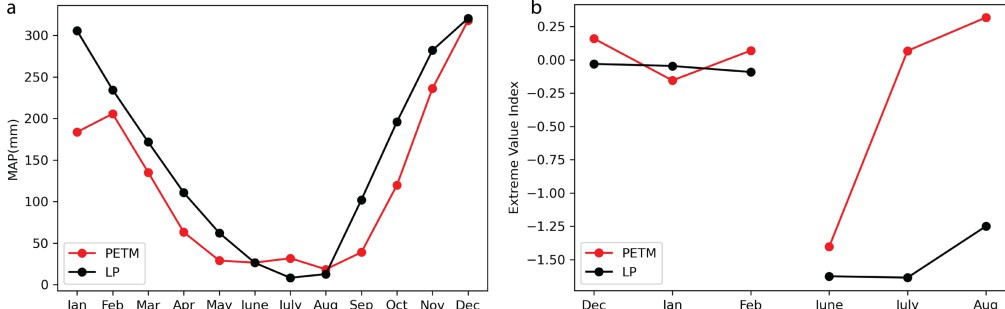


Figure 5. (a) High resolution CAM5 model output of mean monthly precipitation over 15 model
years in central coastal California regions during pre-PETM (Late Paleocene/LP) under low
$pCO_2$ (680 ppmv) and PETM under high $pCO_2$ (1590ppmv). (b) Extreme value index ($\xi$) results
of mean monthly precipitation in winter versus summer of central coastal California region.

**4 Discussion**
Independent proxies (i.e, sediment flux, clay assemblages, and water isotopes ($^2H/^1H$ or $^{18}O/^{16}O$),
though each has limitations, collectively can contribute toward a general picture of how the
mode of precipitation changed (i.e., wetter or dryer, and/or greater seasonality or extremes). For
central California, model simulations of the PETM with low and high resolutions exhibit an
overall decrease in mean annual precipitation (Fig. 4,5). For a mountainous coastal environment
(Fig. 1), siliciclastic sedimentation rates should be highly susceptible to hydrologic changes
among other factors. In addition to relief and lithology, the seasonality of precipitation and
vegetation cover would be major controls on erosion and siliciclastic sediment fluxes. However,
constraining sedimentation rates in shelf sections is challenging given the limitations of



chronostratigraphy in such facies. As such, we rely largely on biostratigraphic constraints with
carbon isotope stratigraphy as the CIE is well captured in both organic and inorganic (calcite)
fossil materials in Lodo (John et al., 2008). This expanded CIE interval indicates higher
sedimentation rates during the PETM (John et al., 2008). Sedimentation rates on the continental
shelf margin are highly sensitive to terrestrial sediment discharge and relative sea-level change.
Assuming the latter was relatively static or rising (Sluijs et al., 2008), the higher sedimentation
rates suggest increasing sediment supply from river runoff to the continental margin, reflecting a
mode shift in regional hydroclimate.

4.1 Hydroclimate response from clay mineralogy
Clay assemblages in nearshore settings can reflect physical and chemical weathering changes as
influenced by regional climate change. Indeed, an increase in the relative abundance of kaolinite
fluxes has been widely observed across the CIE onset in many PETM sections from mid to high
latitudes(Tateo, 2020; Gibson et al., 2000) and interpreted as evidence of a major mode shift in
local hydroclimates. In contrast, the clay mineralogy (Fig. 3) for the Lodo Formation is
dominated mainly by smectite at the onset of the PETM, consistent with seasonal wet/dry cycles
under warm conditions(Gibson et al., 2000). A subtle increase in the kaolinite/smectite hints
slightly enhanced humidity, possibly related to enhanced seasonally subtropical
conditions(Foreman, 2014). Such inferences of hydroclimate changes in the context of coastal
deposition are complicated by fluvial runoff conditions (i.e., provenance, discharge, sediment
influx) and precipitation (i.e., seasonal vs mean annual). Skewed grain size distribution of clay
sediments coinciding with illite/kaolinite peaks (Fig. S1) indicates higher fluvial velocity and
increased erosion as observed in other locations(Chen et al., 2018; Foreman et al., 2012;
Foreman, 2014). The lower Lodo Formation, with relative coarse sandy size distribution
preceding the PETM, shows a slight pulse of kaolinite associated with other minerals, possibly
indicating an early change of hydrological condition in the latest Paleocene before the PETM as
observed elsewhere (Rush et al., 2021). The CIE onset likely represents a transient response to
warming-induced hydroclimate changes, whereas the pre-PETM shift as well as minor variations
post-CIE onset are likely orbitally forced (Kiehl et al., 2018).
4.2 Hydroclimate response from earth system simulations



In all model simulations forced with higher pCO2 (e.g., 3x to 6x pre-industrial), the hydrological
cycle during PETM intensifies as manifested by increases in global mean precipitation and
meridional vapor transport (Kiehl and Shields, 2013; Kiehl et al., 2018; Carmichael et al., 2016;
Zhu et al., 2020). Regionally however, the magnitude and even the sign of precipitation change
can differ considerably from global means. This is most evident in the latest low and high-
resolution model simulations of the PETM. For central California, the simulations yield modest
changes in mean annual precipitation but significant seasonal shifts with a notable decline in
winter precipitation and a slight increase in summer (Fig. 4). This pattern is produced by both the
water isotope enabled iCESM1.2 and the higher resolution CAM5 with an overall shift into
lower amplitude seasonal cycles as a drier winter/spring and a slightly wetter summer (Fig. 4, 5).
This seasonal wet-dry shift appears to be driven in part by a pronounced northward shift of
atmospheric rivers (ARs) in winter along the Pacific coast(Shields et al., 2021). Given the ARs
delivering the majority of winter precipitation to the mid-latitude Pacific coast, less frequent AR
occurrences would result in relatively drier winters during PETM. Moreover, the extreme value
index ($\xi$) shows a small but statistically robust increase winter (DF) wet extremes with a
significant increase in the probability of summer (JJA) wet exceedance during PETM (Fig. 5b).
Although AR related coastal winter storms reduced in frequency (Shields et al., 2021), summer
precipitation increased in intensity thus potentially enhancing individual extremes in this region,
possibly related to an increase in summer tropical storm activity along the Pacific coast during
PETM (Fig. S4).
4.3 $^2$H/$^1$H composition of leaf waxes.
Terrestrial archives exhibit considerable evidence of environmental response to intensified
hydrological cycle during the PETM (McInerney and Wing, 2011). In western North America,
plant fossils show widely expansion (up to 40°N) of tropical rainforest during the PETM along
the east Pacific in mid-latitude(Willis, K.J, McElwain, 2002; Korasidis et al., 2022). Terrestrial
higher plant hydrogen isotope composition (i.e., $\delta^2H_{n\text{-alkane}}$) provide evidence for regional mode
shifts of precipitation (Handley et al., 2008, 2011; Jaramillo et al., 2010; Pagani et al., 2006;
Tipple et al., 2011). Generally, $\delta^2H$ significantly increases in most of these records as expected
with warming, but regionally, notable similarities and differences existed. In subtropical/mid-
latitude regions, $\delta^2H$ increases prior to the PETM followed by a large negative excursion (ca
~20‰) across the onset of PETM (Handley et al., 2008, 2011; Jaramillo et al., 2010; Tipple et





al., 2011). In sharp contrast, high latitudes $\delta^2 H_{n\text{-alkane}}$ show a positive excursion of 55‰ at CIE
onset during PETM, consistent with a reduced meridional temperature gradient and decreasing
isotope distillation during vapor transport (Pagani et al., 2006). However, the Lodo $\delta^2 H_{n\text{-alkane}}$
displays a comparatively muted response, initially decreasing by 25‰ just prior to the CIE onset
followed by a slight $^2H$ enrichment in the main body PETM followed by several shifts toward
more negative values (Fig. 3c). The shift to more negative $\delta^2 H_{n\text{-alkane}}$ values prior to the onset of
CIE in Lodo likely represents background variability related to orbital forcing.
Inferring and comparing the hydroclimate response from leaf water $\delta^2 H$ at any location can be
complicated. Hydrogen isotope fractionation in plants is tightly related to photosynthetic
pathways, source water availability, and atmospheric humidity (Sachse et al., 2012; Tipple et al.,
2015). Along the west coast of North America, no detailed records of vegetation response have
been generated for the PETM. Lack of knowledge about vegetation changes limits our ability to
compute rainwater $\delta^2 H$. Further, under higher weathering rates during the PETM, deep
weathering of Paleocene n-alkanes (Tipple et al., 2011) would possibly dampen of isotopic n-
alkane signals deposited at Lodo. Nevertheless, if we assume the $\delta^2 H_{n\text{-alkane}}$ record reflects only
on changes in source water, the observed modest change of $\delta^2 H_{n\text{-alkane}}$ values at Lodo could be
interpreted in several ways with respect to T-related changes on isotope fractionation offset by
changes in dominant season of precipitation, and/or vapor sources and distance of transport. For
example, a shift in precipitation seasonally between winter and late summer/fall could offset the
effects of warming assuming a shift from a proximal (north or central Pacific) to a more distal
(Gulf of Mexico) source of vapor (Hu and Dominguez, 2015). At ground level, stronger
evapotranspiration during biosynthesis can isotopically be offset by external water source
availability (i.e. seasonal precipitation). Local/regional ground water table variations caused by
hydrological change would also affect the source water-use efficiency of plants since surface
water tends to be more depleted in some perennial species after intense storms in the
groundwater (Hou et al., 2008; Krishnan et al., 2014). Hydrogen isotope fractionation in plants
can also be biased by seasonal shift in regional vegetation growth regime. For example, leaf wax
lipids from terrestrial plants usually record hydrological conditions earlier in the season rather
than fully integrating the entire growing season (Hou et al., 2008; Tipple et al., 2013). Finally,
episodic extremes in precipitation may dominate the hydrogen isotopic composition of the leaf
wax (Krishnan et al., 2014). If most soil water from extreme events during the growth season, the



Lodo $\delta^2H_{n\text{-alkane}}$ changes might reflect a combination effect of intensified seasonal storms with
more $^2$H-depleted precipitation offset by warming induced $^2$H-enrichment in leaf water.

In iCESM1.2 simulations with increasing $p$CO$_2$ (i.e., 3x to 6x pre-industry) and SST, the
seasonal shifts in $\delta^2$H of mean monthly precipitation from pre-PETM to PETM is significant.
Regional $\delta^2H_{prep}$ increases by 10‰ during wet winter while decreasing by ~1 to 5‰ during late
summer/fall in central California (Fig. 4). To estimate how this seasonal change of $\delta^2H_{prep}$ and
precipitation amount influences leaf water $\delta^2$H, we apply a leaf wax proxy model (supplemental
information) which computes the combined effects of changes in seasonal precipitation and
growing season length. The model shows leaf water $\delta^2$H enriched ca. 4 to 7‰ from pre-PETM to
PETM consistent with $\delta^2H_{n\text{-alkane}}$ proxy record in Lodo section. Therefore, this relative muted
leaf wax $\delta^2H_{n\text{-alkane}}$ response can be potentially explained by a seasonal shift of heavy
precipitation events. Alternatively, the change in leaf water $\delta^2$H may also reflect source water
shift of a mixing endmember between proximal and distal sources of water in the coast (Romero
and Feakins, 2011). For example, with a summer shift of source water from the Pacific to
subtropics (i.e., summer monsoons), the effect of increasing distance and distillation would
isotopically deplete vapor (Hu and Dominguez, 2015), thus offsetting the temperature related
enrichment of local $\delta^2H_{precip}$. Infrequent but high intensity tropical cyclones-induced heavy
rainfall in the summer in mid-Pacific during PETM (Kiehl et al., 2021) can also bring the
precipitation more depleted in hydrogen isotope (i.e., a more negative $\delta^2$H).

Finally, a related record that might indirectly reflect on precipitation amount (i.e., atmospheric
humidity) is the n-alkane $\delta^{13}$C and magnitude of the CIE. Recalcitrant higher plants leaf wax
carbon isotope ratios of long-chain n-alkane (n>25 with odd-over-even preference) reflect
mainly carbon source (Diefendorf et al., 2010). However, photosynthetic carbon isotope
fractionation ($\Delta_p$) is sensitive to atmospheric pCO$_2$ variations, generally increase with rising
concentrations assuming a constant photosynthetic fractionation factor and humidity (Diefendorf
et al., 2010). The $\delta^{13}C_{n\text{-alkane}}$ of Lodo section displays a sharp negative shift of ca. 4 ‰ (average
of n-C$_{27}$, n-C$_{29}$, n-C$_{31}$) across the onset of CIE (Fig. 3b), which is consistent with global mean
atmospheric CIE (Sluijs and Dickens, 2012) but generally smaller than observed in other leaf
wax records (Handley et al., 2008, 2011; Jaramillo et al., 2010; Pagani et al., 2006; Tipple et al.,



2011). The smaller $\delta^{13}C_{n-alkane}$ CIE recorded in Lodo could reflect on reduction in local humidity
which preferentially tends to reduce the magnitude of $\Delta_p$ during photosynthetic carbon fixation.

**5 Conclusion**
With PETM greenhouse gas forcing (~56 Ma), climate simulations show an overall decrease in
winter precipitation along the central California margin due in part to a reduction in AR
frequency(Shields et al., 2021), whereas summer precipitation increases slightly. This is
generally consistent with the observations from Lodo Gulch Section based on various
sedimentological and geochemical records, and thus would support a modest reduction in
precipitation (i.e. MAP) along with the possibility of an increase magnitude of extreme
precipitation events during the PETM. In this regard, the observed hydroclimate response during
the PETM as simulated in climate models in response to a doubling (or more) of $CO_2$ could serve
as a past analog for potential hydroclimate changes in California.
Data availability. Data will be available via the PANGAEA repository.
Author contribution. JCZ conceived the project idea, acquired funding and provided overall
supervision. XZ, BJT, WDR, JBN conducted experiments and analyzed the results. JZ, CAS
provide technical expertise in model simulations. All authors contributed to the review and
editing of the manuscript.
Competing interests. The authors declare that they have no conflict of interest.
Acknowledgements.
Funding for this project has been provided by National Science Foundation No. OCE 2103513.
All compound specific isotope analyses were performed at the Yale Institute for Biospheric
Studies-Earth Systems Center for Stable Isotopic Studies that was supported by National Science
Foundation Grant EAR 0628358 and OCE 0902993. The CESM project is supported primarily
by the National Science Foundation (NSF). This material is based upon work supported by the
National Center for Atmospheric Research, which is a major facility sponsored by the NSF under
Cooperative Agreement No. 1852977.




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
