# Peer review of "Response of Coastal California Hydroclimate to the Paleocene- # 2 Eocene Thermal Maximum"

_Climate of the Past, 2023_

## Referee Comment (RC1)

In this paper, Zhang et al. applies a proxy- and model-based approach to reconstruct changes in the local hydrology of central California, from pre-PETM to the PETM. This work builds on data published from a previous study ($\delta^{13}C_{org}$; John et al., 2008) as well as contributing new proxy records (e.g. grain size analyses; clay assemblage analyses; $\delta^{13}C_{n\text{-alkane}}$; and $\delta^2H_{n\text{-alkane}}$). Climate model simulations further support the proxy-based findings and were additionally utilised to constrain the effects of seasonal precipitation on $\delta^2H_{n\text{-alkane}}$ values. They conclude that both the models and proxies indicate an overall drier central California, although the summer saw a slight increase in precipitation. Results from extreme events analyses suggests that intense rainfall events were more frequent during both the winter and the summer.

This paper is well written, containing very little spelling and/or grammatical mistakes. The introduction nicely outlines the significance and the key question that was being investigated. Multiple proxies are utilised in conjunction with a novel method employing models to improve proxy-based reconstructions. The findings address the relevant gaps in our knowledge regarding how the hydrological cycle in central California may respond to future warming. It is exciting to see another study that applies *n*-alkanes as a hydrology proxy, especially as there are only currently seven records for the PETM (Carmichael et al., 2017).

I believe the author can improve on the manuscript by refining the structure of the text. Specifically in regards to the discussion, in addition to creating more continuity between the main text and supplementary information (i.e., removing repetition) (See Section 3). Furthermore, there are a few major questions pertaining to how some of the $\delta^2H_{n\text{-alkane}}$ record has been interpreted (See Section 1).

**1) Interpreting the $\delta^2H_{n\text{-alkane}}$ record**

**1.1 Orbitally driven shift in pre-PETM $\delta^2H_{n\text{-alkane}}$?**

The author states that the 25‰ negative excursion in $\delta^2H_{n\text{-alkane}}$ record, just prior to the onset, is likely representative of orbitally forced variability (Line 303-304). Although this is just a brief sentence and not the focal point of the discussion, I am curious as to what the author based this on. Was there any spectral analyses done to see if the fluctuations in the $\delta^2H_{n\text{-alkane}}$ could correspond to any astronomical forcings? Could the author cite any papers that have looked into potential cyclicity in the hydrological cycle during the Paleogene (e.g., Campbell et al., 2023)? What does this interpretation mean for other sites? The author noted that several subtropical/mid-latitude sites have shown a similar magnitude (~20‰ ) negative shift at the onset of the PETM (e.g., Handley et al., 2008; Jaramillo et al., 2010). How can we go about deconvolving whether such trends are driven by the abrupt perturbations in temperature at the onset of the PETM vs. changes in orbital parameters?

**1.2 Stable $\delta^2H_{n\text{-alkane}}$ through the PETM?**

The results section states that the $\delta^2H_{n\text{-alkane}}$ are relatively invariable throughout the PETM (Line 178-179). Although the PETM is not defined in the figures, if assuming that the PETM includes the CIE up to 20 (unsure depth unit as not defined in figures), Figure 3 presents relative stability at the beginning of the CIE, yet the upper CIE shows larger variability. There is one very negative value at the onset, however, this is one data point and seems to be only with the $C_{29}$ *n*-alkane. On the other hand, the variability in later in the section shows correlation between all the chain lengths and more than one data point. The discussion section largely focuses on explaining the reasons why the record is stable. I was wondering if the author could also touch on why the upper record is more variable. Several other

sites show such variability, for example, TDP Site 14 exhibits oscillations throughout the PETM although the frequency is higher and the magnitude of change lower (Handley et al., 2008).

**1.3 Evidence for a stable hydrological cycle during the beginning of the PETM?**

Although the author describes all the potential factors that may have muted any changes in $\delta^2H_{n\text{-alkane}}$ (i.e., changes in temperature on fractionation vs. the source of water), I was curious as to how they ruled out the simplest explanation that the hydrological cycle may have been stable during the main body of the PETM? Is it because the models and published proxy records suggest the opposite, i.e., higher frequency of extreme rainfall events (Carmichael et al., 2016, 2017). If so, could the author add a sentence to rule out that the lack of change in the $\delta^2H_{n\text{-alkane}}$ record is reflecting the climate, then go on to discuss the other potential explanations.

**2) Utilising *n*-alkane distributions to help interpret the $\delta^2H_{n\text{-alkane}}$ record**

The discussion section mentions that the lack of knowledge on vegetation changes through time hinders the ability to calculate the $\delta^2H$ of precipitation (Line 308-310). I think the lack of change in the average chain length (ACL) is very much worth mentioning here and fits well with the Korasidis et al. (2022) paper, which also shows little change in the Koppen-Geiger climate type within the central California region. There are limitations to using ACL as an indicator for vegetation type (Bush and McInerney, 2013), but it provides some evidence that suggests that the effects of varying fractionation (caused by changing plant types) may have been minimal. With the ACL indicating a mostly terrestrial higher-plant source for the *n*-alkanes, the comment on plant types recording hydrological conditions at a specific season (Line 324-326) can also be of a lesser concern. Even with a strong seasonal signal, if this remained constant throughout the record then the relative changes would be unaffected.

Line 310-312 highlights that the $\delta^2H_{n\text{-alkane}}$ values may be influenced by re-worked *n*-alkanes. I suggest that the author look into the carbon preference index (CPI; Bray and Evans, 1961). This would not require too much work as the author already has *n*-alkane abundance data. The CPI may help indicate any input of thermally mature older sediments/*n*-alkane. CPI values >3–30 would suggest that most of the organic matter is unaltered (Diefendorf and Freimuth, 2017). Furthermore, several studies have suggested input of thermally mature material based on an antiphase between the $\delta^{13}C$ of bulk organic vs. bulk carbonate (e.g., Lyons et al., 2019). If neither of these indicates re-worked *n*-alkanes, this may be highlighted as less of a concern.

**3) Improving the structure**

**3.1 Structure of the methods section and supplementary information**

There is repetition between the methods section in the main manuscript and supplementary information. In addition, there are information that is found in the main manuscript but not the supplementary information and vice versa. For example, it would be useful to have information on how many samples were analysed in Section 2.2.2, instead of noting the instrument used for analyses in both. Similarly, Section 2.2.4 contains a lot of detail that is in the supplementary information, but urea adduction is only in the main manuscript and the column chromatography method is only in the supplementary information. This means that unless the reader looks through both the manuscript and the supplementary information, they are not getting the full picture.

Furthermore, there are no references to the supplementary information in the methods or the results/discussion for the additional figures. This is a minor comment but if the subheadings were labelled in the supplementary information and ordered in a similar way to the main manuscript (i.e.,

leaf wax n-alkane extraction and separation – grain size analyses – extreme value analyses - leaf wax proxy model), then it may be easier to refer to for additional information.

**3.2 Structure of the discussion section**

The first paragraph of the discussion states how sedimentation rates may provide information on the hydrological cycle. Since this study does not present new constraints on the age model or sedimentation rates, I wonder if this could be incorporated into a couple of sentences within the 4.1 section. The crucial point is that higher sedimentation rates suggest more runoff and therefore more rainfall. It would also be interesting to compare the timing of the shift to higher sedimentation rates with the changes in the clay assemblages. The caveats surrounding the lack of tie-points can be raised, but is already discussed in John et al. (2008) and not so much linked to the main proxies within this study.

The discussion paragraphs begin with an introduction to the other studies that have used the same proxy, then highlight the caveats and main assumptions that have to be made. By starting with the issues, the subsequent discussion on the authors results is somewhat downplayed. I personally think that starting with the key findings of this study, then seeing how that compares to other published findings, and then discussing the caveats may flow better. This applies for the paragraph on sedimentation rates but also the paragraph beginning on Line 288 vs. the paragraph beginning on Line 332. Much of the suggestions for why the $\delta^2 H_{n\text{-alkane}}$ values might be muted feel speculative in the first paragraph, however from Line 332 there are really nice evidential based explanations that could be discussed first then the other potential ideas after. In addition, since there is one sentence in the first paragraph (Line 231-232) pertaining to the modelling results, would it make sense to first discuss the modelling results then how the proxies compare to them? However, most of the suggestions on structural changes are based on a subjective preference, so please consider these comments as so.

**Minor comments:**

Line 28: the sentence beginning with "indeed" sounds like it should be related to the previous point, however I would argue that they are two separate and important points. In addition, I think there should be a "the" for "just over the last few decades"…

Line 35: this may be my misunderstanding of what defines a "drought", but is it repetition to say "extreme droughts" and "longer precipitation deficits"?

Line 47-52: for ease of the reader finding the relevant literature, could the citations on Line 48-49 be put next to the relevant locality?

Line 55-57: Cramwinckel et al. (2023) also looks into this. Might be a citation to add here

Line 74: missing a comma after "Here"

Line 76 (plus many other locations): Most often "*n*-alkane" is found italicised

Line 99: how many samples of the originally collected were analysed?

Line 100: how many new samples were collected and also how many of these were analysed

Line 100: good place to note the acronym of bulk sediment organic carbon isotope ($\delta^{13}C_{org}$), and change "analyses" to with an "e" to make it plural

Line 103: spell out iRMS

Line 107: "analyses"

Line 108: how many samples were analysed for grain size analyses?

Line 112: "analyses"

Line 113: should "Kemp et al., 2016" be in brackets?

Line 118: what temperature were the samples dried in?

Line 120: "analyses"

Line 122: could the clay species be specified here?

Line 126: how many samples were analysed for biomarker work? And how many for the CSIA?

Line 127: *v/v* should be italicised

Line 129-130: "Normal-alkanes" can be "*n*-alkanes"

Line 136: numbers by C should be a subscript, e.g., $C_{29}$

Line 143: "analyses"

Line 147 and 141: missing "i" in front of iCESM 1.2?

Line 163: CIE can be abbreviated here rather than on Line 165. How big is the CIE (value ‰). Overall, the results could do with more values replacing descriptive words such as "slight" increase etc.

Line 165: The sentence beginning here and beginning on Line 166 can probably be merged into one sentence, but please cite the other records that are being referred to on Line 166

Line 169: is there a reason why the $\delta^{13}C_{org}$ was plotted in a separate figure to $\delta^{13}C_{n\text{-alkanes}}$? A figure with the bulk and biomarker based isotope records and a figure with the clay assemblage results may work better with the flow of the text

Line 188: how much does the illite/smectite (extra l in illite) and chlorite/smectite ratio increase by?

Line 198-199: how much does the monthly precipitation decrease by and how much is the increase in the summer?

Line 228: different labelling with ($^2H/^1H$ and $^{18}O/^{16}O$) may confuse some readers

Line 232-235: the two sentences here could be merged

Line 253-255: when was the increase in kaolinite/smectite (can be specified in terms of relative to PETM or depth)

Line 257-259: again when was this?

Line 265: similar to comment 1.1 (above) what is the suggestion of orbitally forced variations in the clay assemblage pre-PETM based on?

Line 272: is there a citation for this?

Line 287: $^2H/^1H$ can be used, however, it is nice to remain consistent with naming, i.e., $^2H/^1H$ or $\delta^2H$

Line 312-315: quite a long sentence which makes it hard to follow

Line 318-320: is there a citation for this?

Line 321: I think "affect" should be "effect"?

Line 328: Add "is" to "If most soil water is from…"

Line 333: could the word "significant" be replaced?

Line 335: This is the first time $\delta^2H_{precip}$ is being used. Could this be defined earlier and used throughout both the main text and supplementary information? (assuming the plant is always sourced by precipitation). Also, the "i" is missing in "precip"

Line 351-352: starting the sentence with "higher plants leaf wax" and then saying "long-chain n-alkane" is repetitive. This sentence could remove one and it would still make sense.

Figures:

Figure 1: Is it possible to make the site location more eye-catching by making the red spot larger?

Figure 2: Unit is missing on depth scale (meters?) and some of the clay ratios. Furthermore, could the PETM be highlighted, in addition to using a different symbol to show the already published $\delta^{13}C_{org}$ data (this should also be cited in the caption).

Figure 3: Unit is missing on depth scale. Could the PETM be highlighted? In addition, the caption says "Marine $\delta^{13}C$" (Line 182) which suggests $\delta^{13}C_{carbonate}$ but this is not plotted here. The "n-" can be removed from Line 184.

Figure 5: different labelling to figure 4 (PETM vs. 6x and LP vs. 3x). Is this because they are differently defined? If not, could the same labelling be used for continuity?

Supplementary materials:

Subheadings with analysis should be analyses with an "e" to make it plural.

Most often "*n*-alkane" is found italicised (3 places in supplementary materials).

Second paragraph on "Grain size analysis" section – missing a comma after "and thus hydroclimate during the PETM".

First sentence of "Leaf wax proxy model" – "affect" should be "effect". This sentence currently reads to me like seasonal precipitation effects the fractionation process in plants. Unless this is what is intended, could the sentence be reworded to make clearer what is being discussed, e.g., "To investigate how seasonal variations in the $\delta^2H$ of precipitation effects $\delta^2H_{n\text{-}alkanes}$ values…" or something of that nature.

Fourth paragraph of "Leaf wax proxy model" – could the author cite the paper for which the average chain length equation was taken from or note the equation used in the supplementary information?

First paragraph of "Leaf wax n-alkane extraction and separation" – methanol can be shortened to MeOH as done for DCM. Further in the paragraph, when describing the amount of solvent used during column chromatography, these acronyms can be utilised again. Also "Normal-alkane" x2 can just be labelled as "*n*-alkane"

Third paragraph of "Leaf wax proxy model" – could specify that it is for compound specific analyses.

Fig. S1. Could the boxes be labelled a,b,c etc. to make the caption easier to follow? Also highlight the PETM in the lower most plot.

**References**

Campbell, J., Poulsen, C.J., Zhu, J., Tierney, J.E. and Keeler, J., 2023. CO 2-and orbitally-driven oxygen isotope variability in the Early Eocene. *Climate of the Past Discussions*, *2023*, pp.1-32.

Diefendorf, A.F. and Freimuth, E.J., 2017. Extracting the most from terrestrial plant-derived n-alkyl lipids and their carbon isotopes from the sedimentary record: A review. *Organic Geochemistry*, *103*, pp.1-21.

Lyons, S.L., Baczynski, A.A., Babila, T.L., Bralower, T.J., Hajek, E.A., Kump, L.R., Polites, E.G., Self-Trail, J.M., Trampush, S.M., Vornlocher, J.R. and Zachos, J.C., 2019. Palaeocene–Eocene thermal maximum prolonged by fossil carbon oxidation. *Nature Geoscience*, *12*(1), pp.54-60.

**Signed** Emily H Hollingsworth

---

## Referee Comment (RC2)

Review

In this paper, Zhang et al present a multi-proxy and multi-model investigation of hydrological change at the PETM on the western coast of North America. This is a topic that intrigues nearly everyone studying ancient greenhouse climates and the work represents a compelling variety of methods. Collectively, these provide new insights into the nature of regional (and global) hydrometeorology, including evidence for changes in seasonality and extreme rainfall events. It builds on previous work in exciting ways, especially via the data-proxy comparison. It certainly should be published.

However, I do think the paper could be significantly improved. The data and discussion are sometimes presented too briefly and the interpretations are somewhat unclear. Often a range of explanations are offered (which I appreciate) but with no effort to distinguish them or to make use of the multi-proxy data to integrate them. Overall, I think the interpretations are robust and the caveats duly noted, but the reasoning is not always clearly laid out or explained. Other potentially interesting data are ignored (esp post-CIE data or n-alkane distributions). All of that probably sounds rather critical, but I do not want to discourage the authors! The work done is impressive; I think this is a perfectly adequate paper – but there is probably a more exciting paper that better utilises all of that work.

Below are some comments that hopefully elaborate those thoughts in a constructive way (NOTE that some of these are quite important recommendations whereas others are suggestions and I trust the authors and editor to distinguish the two; but happy to be contacted offline if it is unclear):

Abstract and lines 34-42 of the introduction and lines 371-372 of conclusions: I am not convinced that the framing around California's current or future hydroclimate is necessary or appropriate. It is one thing to treat the PETM as an analogue for the future and another to treat Lodo Gulch as an analogue for California's future. As the authors note, understanding regional responses to warming is essential, and I would focus more on that framing.

Figure 1 is certainly adequate but I think it could be strengthened by linking it to previous studies, i.e. including adding Big Horn Basin sites, and by adding in some atmospheric circulation features that are discussed frequently in the paper.

Line 98: Maybe specify 'bulk organic stable carbon isotopes' in the title for clarity. (And align that title with section 3.1).

Lines 118-119: There is some shifting from past to present tense. It would be useful to check this throughout the Methods.

Line 124: I suggest retitling this as 'Leaf Wax distributions and carbon and hydrogen isotopic compositions.' Note that there are a few chemistry conventions that should be properly sorted – n should be italicized in $n$-alkane and carbon numbers should be subscripts in line 136 (And check throughout the manuscript). In line 140, 'were' should replace 'was.'

Lines 142-159: I am delighted to see the proxy data compared to isotope enabled models. That is a strength of this paper. However, hydrological processes are notoriously variable amongst climate models. It would be useful to briefly draw on DeepMIP (or similar) studies to summarise how CESM compares to other models. Is it 'typical', an 'outlier', etc? This could be a whole paper in itself and I certainly am not suggesting the authors add extensive text, but only enough text to help readers put these findings into context.

The authors should consider flipping the order of figures 2 and 3 to better align with the text.

Lines 162 to 179: I trust the authors, but please include n-alkane CPIs and TARs in the figures and a chromatogram (and proportional abundances) in the SI so that we can be confident that the n-alkanes have a leaf wax distribution. And explain and justify that in the text.

Lines 163 to 168: It might be worth noting that the CIE recorded by the n-alkanes is larger than that recorded by bulk organic matter (as is observed in other records), but also that the bulk d13C values never return to pre-CIE values. Also, the authors write that the top of the PETM body is marked by the truncation of the n-alkane CIE; presumably that means they trust it more than the bulk organic CIE? Also, could the PETM body not have been truncated earlier? And the truncation could also include not just the PETM body but the return. Finally, no information is given on the NP biozones. Without overly reproducing the info in John et al., it would be useful to add a few sentences on the stratigraphy, the uncertainty, the age gap, etc and to label the inferred PETM interval explicitly on the figure. (This will also help with subsequent sections, such as lines 236-237, where the authors discuss the challenges of determining sedimentation rates).

Lines 177-179: I don't think that Results sections should be excessive, but this is a bit perfunctory. The brief negative spikes are very large and merit a few more words, especially as one of those appears to be in a coarser lithology than the other data (and is it 'one' or 'two' brief intervals?). Also, some of those negative values appear to post-date the PETM body (see previous comment) so it is worth describing the stratigraphic occurrence of these data with greater precision. They largely ignore these negative spikes in the discussion and I suspect that could be justified by a more thorough Results section.

The 'slight enrichment' in the main body seems very slight indeed and at the limit of analytical error (6‰; line 140). The negative shift prior to the PETM is recorded by only two pre-PETM data points and that should be acknowledged. Perhaps even more important is the fact that post-PETM d2H values are 2H-enriched relative to those two samples but similar to those of the PETM.

Lines 186 to 191. Great to see clay mineralogy woven into this study. Like my comments on d2H values, this section would benefit from some expansion. In particular, I would note that many of the clay mineral assemblages – especially and intriguingly kaolinite to smectite - never return to pre-PETM values (Although our record is more limited, we see aspects of this at Tanzania as well).

Figure 4: For the published version, please make the text larger and edit the text in the figure caption (there are a number of typos).

Figure 5: Why not show the extreme value index for all months (just for completeness)?

Lines 237 to 239: This text confused me a bit. First, the authors really have not constrained the PETM in the previous text. Second, the CIE thickness (if complete) does not allow for determination of the change in sedimentation rates. I think the authors are trying to briefly explain what John et al. (2008) determined, but as written that is unclear. I think this opening paragraph would be stronger if it clearly explained what has been determined previously in this region and by whom, and then ended with a clear list of how the subsequent discussion sections are going to elaborate on that understanding.

Discussion: A general observation of the discussion is that it treats the data in rather isolated silos (with the exception of using models to interpret leaf wax d2H values). And both sections 4.1 and 4.2 seem less like discussions than extensions of the associated Results. Since the Results have already been presented, then draw from all of them to drive the discussion forward. For example, I would not have a discussion section on clay mineralogy but rather one on extreme rainfall events that

draws on the mineralogy and the models. That is just a suggestion, of course, and I am one to give authors latitude in how they want to tell their story! But I think a more integrated approach would ensure that the greatest added value emerges from the multi-proxy study.

Lines 255-257: I don't think these comments quite capture the debate about clay mineralogical change at the PETM. The increase in kaolinite has been attributed to both increased humidity and more deeply erosive events; given the context of the paper, I would make those two interpretations explicit. And then… is there any evidence to distinguish between those? The model simulations (or at least what is included) suggests that extreme events and erosion are more likely explanations than increased humidity. If so, say that. Also, I'd encourage the authors to discuss the post-PETM data and allow that to inform their interpretation.

Lines 301 to 303: See comments above – the description of these records needs to recognize the analytical error and be presented with a wee bit better stratigraphic rigour.

Lines 303 to 304: Ascribing the shift in d2H prior to the PETM to orbital variability seems bold. What is the evidence for this? And why don't we see similar orbital variability during the PETM? Or afterwards?

Paragraph starting line 305: This would be easier to follow if the authors clearly set out what behaviour they are attempting to interpret. I assume (but am not sure) that they are arguing that leaf wax d2H – and by extension local meteoric water, given the caveats they correctly note – does not change much in their record (barring a few anomalies). State that clearly. It will make the rest of the text easier to write and to follow. For example, it will allow the reader to understand why we are discussing different factors that could 'offset'.

In addition, I feel like the authors have said that 'we have some data and there are a lot of explanations for it' without drawing on other data to try to narrow down and distinguish hypotheses. What does the mineralogy say about changes in precipitation? What do n-alkane CPIs say about reworked OM? What do ACLs (in SI but never mentioned) say about changes in vegetation? A stronger structure and a more comprehensive discussion will allow more compelling interpretations.

Lines 338-339: This is a really nice application of the model. But the data are not convincing. I am not convinced that there is an analytically significant shift across the PETM (see line 140). And I certainly don't think it is significant in the context of the entire record. But there is such a compelling story here! Based on other mid- and low-latitude sites, we expect a strong positive d2H shift. Assuming plants record annual precip d2H, then the authors' models also predict that. The fact that this is not seen can be resolved by considering a change in seasonal precipitation d2H and growth. That approach predicts a leaf wax d2H shift that is very small and likely below analytical error, and that is what is observed. That is a really nice finding.

(In fact, it is so nice that I'd like to see the authors validate it a bit – perhaps in the SI by determining if the models can predict leaf wax d2H changes at other sites. If the same approach that yields a minor shift in California also yields a minor shift in Europe and a strong positive shift in Tanzania, then that is very compelling. There has been a big opportunity missed by not using the model to assess global d2H records. Maybe for a future paper…)

Lines 350: I like this inclusion of the d13C record.

End Discussion and Conclusion: The authors have a nice integrated dataset. But they never quite draw it all together into a holistic picture. For example, the d13C record is used to infer lower

humidity, but that is not mentioned in the abstract or conclusions.  The conclusions mention lower winter precip and slightly higher summer precip but do not make it clear that the overall annual precipitation is much lower in the 6x CO2 simulation.  Picking through all of the data, it seems that there is evidence for decreased overall precipitation, especially in the winter; that the precip also becomes more episodic; that these factors and higher temperatures have combined to yield a more arid climate and that impacted the vegetation as expressed in d13C values.  All of these will have contributed to a more erosive sedimentary regime.  These interpretations are validated by leaf wax d2H values – but that could only be deduced with careful data-model comparison that allowed the competing controls on plant d2H to be constrained. This is a really interesting suite of data, but it does not quite come together as it could.

 The paper has a fairly large number of grammatical errors that should be cleaned up on editing.  I caught several in the abstract, but they generally become more common further into the manuscript.  There are many of them in some figure captions.

---

## Author Comment (AC1)

Reply to review 1:

Emily H Hollingsworth (Referee)

**The authors thank the reviewer for the positive feedback and constructive suggestions. We have addressed most of the comments and a point-by-point reply is provided below.**

*In this paper, Zhang et al. applies a proxy- and model-based approach to reconstruct changes in the local hydrology of central California, from pre-PETM to the PETM. This work builds on data published from a previous study ($\delta^{13}Corg$; John et al., 2008) as well as contributing new proxy records (e.g. grain size analyses; clay assemblage analyses; $\delta^{13}Cn$-alkane; and $\delta^2Hn$-alkane). Climate model simulations further support the proxy-based findings and were additionally utilised to constrain the effects of seasonal precipitation on $\delta^2Hn$-alkane values. They conclude that both the models and proxies indicate an overall drier central California, although the summer saw a slight increase in precipitation. Results from extreme events analyses suggests that intense rainfall events were more frequent during both the winter and the summer.*

*This paper is well written, containing very little spelling and/or grammatical mistakes. The introduction nicely outlines the significance and the key question that was being investigated. Multiple proxies are utilised in conjunction with a novel method employing models to improve proxy-based reconstructions. The findings address the relevant gaps in our knowledge regarding how the hydrological cycle in central California may respond to future warming. It is exciting to see another study that applies n-alkanes as a hydrology proxy, especially as there are only currently seven records for the PETM (Carmichael et al., 2017).*

*I believe the author can improve on the manuscript by refining the structure of the text. Specifically in regards to the discussion, in addition to creating more continuity between the main text and supplementary information (i.e., removing repetition) (See Section 3). Furthermore, there are a few major questions pertaining to how some of the $\delta2Hn$-alkane record has been interpreted (See Section 1).*

*1) Interpreting the $\delta^2Hn$-alkane record*

*1.1 Orbitally driven shift in pre-PETM $\delta2Hn$-alkane?*
*The author states that the 25‰ negative excursion in $\delta2Hn$-alkane record, just prior to the onset, is likely representative of orbitally forced variability (Line 303-304). Although this is just a brief sentence and not the focal point of the discussion, I am curious as to what the author based this on. Was there any spectral analyses done to see if the fluctuations in the $\delta^2Hn$-alkane could correspond to any astronomical forcings? Could the author cite any papers that have looked into potential cyclicity in the hydrological cycle during the Paleogene (e.g., Campbell et al., 2023)? What does this interpretation mean for other sites? The author noted that several subtropical/mid-latitude sites have shown a similar magnitude (~20‰) negative shift at the onset of the PETM (e.g., Handley et al., 2008; Jaramillo et al., 2010). How can we go about*

*deconvolving whether such trends are driven by the abrupt perturbations in temperature at the onset of the PETM vs. changes in orbital parameters?*

> This is a fair criticism. The signal is small and at this location there's not a sufficiently long upper Paleocene record to establish the background variability (related to orbital or other forcing) prior to the PETM. We mention orbital forcing simply because there are several sites (e.g. Forada, Tanzania, New Zealand, Venezuela etc.) showing a positive shift prior to PETM (line 296-297) opposite of Lodo. If somehow co-eval in time, opposite patterns would be more consistent with orbital forcing on local precipitation, in part supported by theory (see Kiel et al 2018; Lunt et al., 2007; Rush et al., 2021). Given the poor age control on all these sites, however, it is just as likely that these changes may not be coincident. Sure, the local signal could be related to warming but really difficult to prove either way (w/o constraints on T). As suggested, we added a citation to Campbell et al., 2023**.** The initial enrichment (~5‰) at the onset of the PETM is consistent with simulated response for this region.

1.2 Stable $\delta^2H$ *n*-alkane through the PETM?

The results section states that the $\delta^2H$ *n*-alkane are relatively invariable throughout the PETM (Line 178-179). Although the PETM is not defined in the figures, if assuming that the PETM includes the CIE up to 20 (unsure depth unit as not defined in figures), Figure 3 presents relative stability at the beginning of the CIE, yet the upper CIE shows larger variability. There is one very negative value at the onset, however, this is one data point and seems to be only with the C29 *n*-alkane. On the other hand, the variability in later in the section shows correlation between all the chain lengths and more than one data point. The discussion section largely focuses on explaining the reasons why the record is stable. I was wondering if the author could also touch on why the upper record is more variable. Several other sites show such variability, for example, TDP Site 14 exhibits oscillations throughout the PETM although the frequency is higher and the magnitude of change lower (Handley et al., 2008).

> Yes, overall through the onset and CIE, $\delta^2H$ is relatively stable with just a slight enrichment (as noted above). The CIE recovery interval, now highlighted, is truncated around 22m roughly coincident with an increase in the variability of $\delta^2H_{n-alkane}$. As we have limited age model control, mainly relying on biostratigraphy, it's difficult to interpret the cause of the increased variability without a lot of speculation. Just considering the depositional facies and environment, there's potential for artefacts (e.g., truncation) related to stratigraphic breaks, etc.

*1.3 Evidence for a stable hydrological cycle during the beginning of the PETM?*

*Although the author describes all the potential factors that may have muted any changes in δ2Hn-alkane (i.e., changes in temperature on fractionation vs. the source of water), I was curious as to how they ruled out the simplest explanation that the hydrological cycle may have been stable during the main body of the PETM? Is it because the models and published proxy*

*records suggest the opposite, i.e., higher frequency of extreme rainfall events (Carmichael et al., 2016, 2017). If so, could the author add a sentence to rule out that the lack of change in the δ2Hn-alkane record is reflecting the climate, then go on to discuss the other potential explanations.*

> We do favor the simplest explanation from the observational perspective, no clear pattern of a "major" change in regional hydrology (as compared to other sections), whereas the models suggest a significant reduction in winter precipitation. We modified the text (start line 419) to emphasize the relatively muted response of the leaf wax record.

*2) Utilising n-alkane distributions to help interpret the δ2Hn-alkane record*

*The discussion section mentions that the lack of knowledge on vegetation changes through time hinders the ability to calculate the δ2H of precipitation (Line 308-310). I think the lack of change in the average chain length (ACL) is very much worth mentioning here and fits well with the Korasidis et al. (2022) paper, which also shows little change in the Koppen-Geiger climate type within the central California region. There are limitations to using ACL as an indicator for vegetation type (Bush and McInerney, 2013), but it provides some evidence that suggests that the effects of varying fractionation (caused by changing plant types) may have been minimal. With the ACL indicating a mostly terrestrial higher-plant source for the n-alkanes, the comment on plant types recording hydrological conditions at a specific season (Line 324-326) can also be of a lesser concern. Even with a strong seasonal signal, if this remained constant throughout the record then the relative changes would be unaffected.*
*Line 310-312 highlights that the δ2Hn-alkane values may be influenced by re-worked n-alkanes. I suggest that the author look into the carbon preference index (CPI; Bray and Evans, 1961). This would not require too much work as the author already has n-alkane abundance data. The CPI may help indicate any input of thermally mature older sediments/n-alkane. CPI values >3–30 would suggest that most of the organic matter is unaltered (Diefendorf and Freimuth, 2017). Furthermore, several studies have suggested input of thermally mature material based on an antiphase between the δ13C of bulk organic vs. bulk carbonate (e.g., Lyons et al., 2019). If neither of these indicates re-worked n-alkanes, this may be highlighted as less of a concern.*

> Thanks for the suggestions. We added explanation of the lack of change in ACL and cited Korasidis et al. (2022) to better constrain the effects of varying fractionation caused by vegetation changes. We also added the CPI in the figure to support limited recorking of *n*-alkanes.

*3) Improving the structure*
*3.1 Structure of the methods section and supplementary information*

*There is repetition between the methods section in the main manuscript and supplementary information. In addition, there are information that is found in the main manuscript but not the*

*supplementary information and vice versa. For example, it would be useful to have information on how many samples were analysed in Section 2.2.2, instead of noting the instrument used for analyses in both. Similarly, Section 2.2.4 contains a lot of detail that is in the supplementary information, but urea adduction is only in the main manuscript and the column chromatography method is only in the supplementary information. This means that unless the reader looks through both the manuscript and the supplementary information, they are not getting the full picture.*

> We added the 35 samples analyzed in the main text method section. We removed the sample preparation in the supplementary information to avoid repetition and merged the column chromatography into the main text method section.

*Furthermore, there are no references to the supplementary information in the methods or the results/discussion for the additional figures. This is a minor comment but if the subheadings were labelled in the supplementary information and ordered in a similar way to the main manuscript (i.e.,*
*leaf wax n-alkane extraction and separation – grain size analyses – extreme value analyses - leaf wax proxy model), then it may be easier to refer to for additional information.*

> We added the related references to the supplementary information where appropriate. We reorganized each section in the supplementary information to align with the main text.

*3.2 Structure of the discussion section*

*The first paragraph of the discussion states how sedimentation rates may provide information on the hydrological cycle. Since this study does not present new constraints on the age model or sedimentation rates, I wonder if this could be incorporated into a couple of sentences within the 4.1 section. The crucial point is that higher sedimentation rates suggest more runoff and therefore more rainfall. It would also be interesting to compare the timing of the shift to higher sedimentation rates with the changes in the clay assemblages. The caveats surrounding the lack of tie-points can be raised, but is already discussed in John et al. (2008) and not so much linked to the main proxies within this study.*

> The previous observation of a shift in sedimentation rate is consistent with increased runoff so seemed appropriate to start the discussion with. Given the uncertainties in age control, it sets the stage for discussing the other observation proxies.

*The discussion paragraphs begin with an introduction to the other studies that have used the same proxy, then highlight the caveats and main assumptions that have to be made. By starting with the issues, the subsequent discussion on the authors results is somewhat downplayed. I personally think that starting with the key findings of this study, then seeing how that compares to other published findings, and then discussing the caveats may flow better. This applies for the*

*paragraph on sedimentation rates but also the paragraph beginning on Line 288 vs. the paragraph beginning on Line 332. Much of the suggestions for why the δ2Hn-alkane values might be muted feel speculative in the first paragraph, however from Line 332 there are really nice evidential based explanations that could be discussed first then the other potential ideas after. In addition, since there is one sentence in the first paragraph (Line 231-232) pertaining to the modelling results, would it make sense to first discuss the modelling results then how the proxies compare to them? However, most of the suggestions on structural changes are based on a subjective preference, so please consider these comments as so.*

This is a reasonable suggestion as we struggled a bit with organization of the discussion. We have revised the structure to start with a discussion of the model simulations, followed by the comparison with observations. We believe it now flows more smoothly.

**Minor comments:**
*Line 28: the sentence beginning with "indeed" sounds like it should be related to the previous point, however I would argue that they are two separate and important points. In addition, I think there should be a "the" for "just over the last few decades"*

Done.

*Line 35: this may be my misunderstanding of what defines a "drought", but is it repetition to say "extreme droughts" and "longer precipitation deficits"?*

Done.

*Line 47-52: for ease of the reader finding the relevant literature, could the citations on Line 48-49 be put next to the relevant locality?*

Agreed.

*Line 55-57: Cramwinckel et al. (2023) also looks into this. Might be a citation to add here.*

An oversight. Citation added.

*Line 74: missing a comma after "Here"*

Done.

*Line 76 (plus many other locations): Most often "**n**-alkane" is found italicised*

Done.

*Line 99: how many samples of the originally collected were analysed?*

27 Samples added.

*Line 100: how many new samples were collected and also how many of these were analysed*

Done.

*Line 100: good place to note the acronym of bulk sediment organic carbon isotope (δ13Corg), and change "analyses" to with an "e" to make it plural.*

Done

*Line 107: "analyses"*

Done

*Line 108: how many samples were analysed for grain size analyses?*

39 samples added in the text.

*Line 112: "analyses"*

Done

*Line 113: should "Kemp et al., 2016" be in brackets?*

Changed into Kemp et al., (2016)

*Line 118: what temperature were the samples dried in?*

40 °C added into the text.

*Line 120: "analyses"*
Done

*Line 122: could the clay species be specified here?*

Yes, added.

*Line 126: how many samples were analysed for biomarker work? And how many for the CSIA?*

Added in the text.

*Line 127: v/v should be italicised*
Done

*Line 129-130: "Normal-alkanes" can be "n-alkanes"*
Done

*Line 136: numbers by C should be a subscript, e.g., C29*

Done

*Line 143: "analyses"*
    Done

*Line 147 and 141: missing "i" in front of iCESM 1.2?*

    No i for this model since it is the original Climate Earth System Model. When using isotope-enabled CESM, the acronym can be iCESM.

*Line 163: CIE can be abbreviated here rather than on Line 165. How big is the CIE (value ‰). Overall, the results could do with more values replacing descriptive words such as "slight" increase etc.*
    Done

*Line 165: The sentence beginning here and beginning on Line 166 can probably be merged into one sentence, but please cite the other records that are being referred to on Line 166*

    Done

*Line 169: is there a reason why the δ13Corg was plotted in a separate figure to δ13Cn-alkanes? A figure with the bulk and biomarker based isotope records and a figure with the clay assemblage results may work better with the flow of the text*

    Both fig 2 and fig3 have $\delta^{13}C_{org}$ record.

*Line 188: how much does the illite/smectite (extra l in illite) and chlorite/smectite ratio increase by?*

    The illite/smectite ratio increases from 0.45 (pre-PETM) to 2 (PETM), and chlorite/smectite ratio increases from 0.29 to 1. Numbers have been added in the text.

*Line 198-199: how much does the monthly precipitation decrease by and how much is the increase in the summer?*

    Added in the text.

*Line 228: different labelling with (2H/1H and 18O/16O) may confuse some readers*

    Done. Changed into $\delta^2H$ or $\delta^{18}O$

*Line 232-235: the two sentences here could be merged*

    Done

*Line 253-255: when was the increase in kaolinite/smectite (can be specified in terms of relative to PETM or depth)*

Clarified in the result section 3.3.

*Line 257-259: again when was this?*

Clarified in the result section 3.3.

*Line 265: similar to comment 1.1 (above) what is the suggestion of orbitally forced variations in the clay assemblage pre-PETM based on?*

See reply in 1.1

*Line 272: is there a citation for this?*

Based on the references cited in the model simulations right before the sentence.

*Line 287: 2H/1H can be used, however, it is nice to remain consistent with naming, i.e., 2H/1H or δ2H*

Done. All in $\delta^2$H.

*Line 312-315: quite a long sentence which makes it hard to follow*

Revised.

*Line 318-320: is there a citation for this?*
Added

*Line 321: I think "affect" should be "effect"?*

Prefer to use affect as verb to address the action of influencing.

*Line 328: Add "is" to "If most soil water is from…"*

Done

*Line 333: could the word "significant" be replaced?*

No

*Line 335: This is the first time δ2Hprecip is being used. Could this be defined earlier and used throughout both the main text and supplementary information? (assuming the plant is always sourced by precipitation). Also, the "i" is missing in "precip"*

   Done

*Line 351-352: starting the sentence with "higher plants leaf wax" and then saying "long-chain n-alkane" is repetitive. This sentence could remove one and it would still make sense.*

   Done

Figures:
*Figure 1: Is it possible to make the site location more eye-catching by making the red spot larger?*

   Done

*Figure 2: Unit is missing on depth scale (meters?) and some of the clay ratios. Furthermore, could the PETM be highlighted, in addition to using a different symbol to show the already published δ13Corg data (this should also be cited in the caption).*

   Done

*Figure 3: Unit is missing on depth scale. Could the PETM be highlighted? In addition, the caption says "Marine δ13C" (Line 182) which suggests δ13Ccarbonate but this is not plotted here. The "n-" can be removed from Line 184.*

   Done.

*Figure 5: different labelling to figure 4 (PETM vs. 6x and LP vs. 3x). Is this because they are differently defined? If not, could the same labelling be used for continuity?*

   Yes. It's defined in different climate models.

Supplementary materials:

*Subheadings with analysis should be analyses with an "e" to make it plural.*

   Done

*Most often "**n**-alkane" is found italicised (3 places in supplementary materials).*

     Done

*Second paragraph on "Grain size analysis" section – missing a comma after "and thus hydroclimate during the PETM".*

     Done

*First sentence of "Leaf wax proxy model" – "affect" should be "effect". This sentence currently reads to me like seasonal precipitation effects the fractionation process in plants. Unless this is what is intended, could the sentence be reworded to make clearer what is being discussed, e.g., "To investigate how seasonal variations in the δ2H of precipitation effects δ2Hn-alkanes values…" or something of that nature.*

     Done

*Fourth paragraph of "Leaf wax proxy model" – could the author cite the paper for which the average chain length equation was taken from or note the equation used in the supplementary information?*

     Done

*First paragraph of "Leaf wax n-alkane extraction and separation" – methanol can be shortened to MeOH as done for DCM. Further in the paragraph, when describing the amount of solvent used during column chromatography, these acronyms can be utilised again. Also "Normal-alkane" x2 can just be labelled as "n-alkane"*

     Done

*Third paragraph of "Leaf wax proxy model" – could specify that it is for compound specific analyses.*

     This is a general leaf water proxy model.

*Fig. S1. Could the boxes be labelled a,b,c etc. to make the caption easier to follow? Also highlight the PETM in the lower most plot.*

     Done

---

## Author Comment (AC2)

Response to review 2

Richard Pancost (Referee)

**The authors thank the reviewer for the positive feedback and recommendation for publication. We have adapted most of the suggestions/edits. A point-by-point response is provided below.**

*In this paper, Zhang et al present a multi-proxy and multi-model investigation of hydrological change at the PETM on the western coast of North America. This is a topic that intrigues nearly everyone studying ancient greenhouse climates and the work represents a compelling variety of methods. Collectively, these provide new insights into the nature of regional (and global) hydrometeorology, including evidence for changes in seasonality and extreme rainfall events. It builds on previous work in exciting ways, especially via the data-proxy comparison. It certainly should be published.*

*However, I do think the paper could be significantly improved. The data and discussion are sometimes presented too briefly and the interpretations are somewhat unclear. Often a range of explanations are offered (which I appreciate) but with no effort to distinguish them or to make use of the multi-proxy data to integrate them. Overall, I think the interpretations are robust and the caveats duly noted, but the reasoning is not always clearly laid out or explained. Other potentially interesting data are ignored (esp post-CIE data or n-alkane distributions). All of that probably sounds rather critical, but I do not want to discourage the authors! The work done is impressive; I think this is a perfectly adequate paper – but there is probably a more exciting paper that better utilises all of that work.*

> This is a fair critique. We expanded the results and reorganized the discussion section following several of the suggestions provided above/below.

*Abstract and lines 34-42 of the introduction and lines 371-372 of conclusions: I am not convinced that the framing around California's current or future hydroclimate is necessary or appropriate. It is one thing to treat the PETM as an analogue for the future and another to treat Lodo Gulch as an analogue for California's future. As the authors note, understanding regional responses to warming is essential, and I would focus more on that framing.*

> While we believe the paper is motivated in part (lines 34-42) by California's current/future hydroclimate shifts, particularly the frequency/intensity of AR's, we have condensed that section, eliminating most of the non-relevant details.

Figure 1 is certainly adequate but I think it could be strengthened by linking it to previous studies, i.e. including adding Big Horn Basin sites, and by adding in some atmospheric circulation features that are discussed frequently in the paper.

Added the Big Horn Basin site to the map for reference.

*Line 98: Maybe specify 'bulk organic stable carbon isotopes' in the title for clarity. (And align that title with section 3.1).*

We made the change and aligned in section 3.1 title.

*Lines 118-119: There is some shifting from past to present tense. It would be useful to check this throughout the Methods.*

We revised the grammar tense throughout the paper.

*Line 124: I suggest retitling this as 'Leaf Wax distributions and carbon and hydrogen isotopic compositions.' Note that there are a few chemistry conventions that should be properly sorted – n should be italicized in n-alkane and carbon numbers should be subscripts in line 136 (And check throughout the manuscript). In line 140, 'were' should replace 'was.'*

Have revised notations throughout the manuscript.

*Lines 142-159: I am delighted to see the proxy data compared to isotope enabled models. That is a strength of this paper. However, hydrological processes are notoriously variable amongst climate models. It would be useful to briefly draw on DeepMIP (or similar) studies to summarise how CESM compares to other models. Is it 'typical', an 'outlier', etc? This could be a whole paper in itself and I certainly am not suggesting the authors add extensive text, but only enough text to help readers put these findings into context.*

Added background information of the CESM model simulations within the context of DeepMIP protocols to describe the specific model configurations and strengths over other models on hydrological sensitivity. And cited Cramwinckel (et al., 2022) to clarify that ''crucially, the models with reduced latitudinal temperature gradients (e.g., GFDL, CESM) more closely reproduce proxy-derived precipitation estimates and other key climate metrics''.

*The authors should consider flipping the order of figures 2 and 3 to better align with the text.*

Good suggestion. Have reorganized figures 2 and 3.

*Lines 162 to 179: I trust the authors, but please include n-alkane CPIs and TARs in the figures and a chromatogram (and proportional abundances) in the SI so that we can be confident that the n-alkanes have a leaf wax distribution. And explain and justify that in the text.*

We added the CPIs in the figure 2. We didn't have any aquatic *n*-alkanes in the samples. Histograms representing the distribution of long-chain *n*-alkanes are added to the supplementary information.

*Lines 163 to 168: It might be worth noting that the CIE recorded by the n-alkanes is larger than that recorded by bulk organic matter (as is observed in other records), but also that the bulk d13C values never return to pre-CIE values. Also, the authors write that the top of the PETM body is marked by the truncation of the n-alkane CIE; presumably that means they trust it more than the bulk organic CIE? Also, could the PETM body not have been truncated earlier? And the truncation could also include not just the PETM body but the return. Finally, no information is given on the NP biozones. Without overly reproducing the info in John et al., it would be useful to add a few sentences on the stratigraphy, the uncertainty, the age gap, etc and to label the inferred PETM interval explicitly on the figure. (This will also help with subsequent sections, such as lines 236-237, where the authors discuss the challenges of determining sedimentation rates).*

> The pattern in the bulk $C_{org}$ is noisy and so not very useful for much, other than identifying the onset of the CIE. The *n*-alkane record, however, is much cleaner and so worth discussing. We have modified the description of the results, relation of the abrupt return to the unconformity and NP boundary (10/11) and mention the fact that as observed in most marine sections, the C isotope values doesn't return to pre-excursion values**.**

*Lines 177-179: I don't think that Results sections should be excessive, but this is a bit perfunctory. The brief negative spikes are very large and merit a few more words, especially as one of those appears to be in a coarser lithology than the other data (and is it 'one' or 'two' brief intervals?). Also, some of those negative values appear to post-date the PETM body (see previous comment) so it is worth describing the stratigraphic occurrence of these data with greater precision. They largely ignore these negative spikes in the discussion and I suspect that could be justified by a more thorough Results section.*

*The 'slight enrichment' in the main body seems very slight indeed and at the limit of analytical error (6‰; line 140). The negative shift prior to the PETM is recorded by only two pre-PETM data points and that should be acknowledged. Perhaps even more important is the fact that post-PETM d2H values are 2H-enriched relative to those two samples but similar to those of the PETM.*

> We mention the anomalies and relation to lithologic features (or not). The one large negative anomaly occurs with the disconformity which is related to a sea level regression. The pre-excursion shift is likely related to some regional response, possibly to orbital forcing. Other pre-excursion shifts occur around the world though the sense of change is both positive and negative which would be consistent with orbital forcing on

local precipitation. The bottom line is that the lack of control on background variability associated with orbital forcing cannot be constrained here.

Lines 186 to 191. Great to see clay mineralogy woven into this study. Like my comments on d2H values, this section would benefit from some expansion. In particular, I would note that many of the clay mineral assemblages – especially and intriguingly kaolinite to smectite - never return to pre-PETM values (Although our record is more limited, we see aspects of this at Tanzania as well).

Expanded the description of clay assemblage patterns within and post-PETM.

Figure 4: For the published version, please make the text larger and edit the text in the figure caption (there are a number of typos).

Done.

*Figure 5: Why not show the extreme value index for all months (just for completeness)?*

The purpose of the extreme value index is to compare the seasonal shift of extreme precipitation probability from pre-PETM to PETM, so we want to highlight the seasonal contrast from two seasons have most and least precipitations. For completeness, we added the other months in the supplemental information.

*Lines 237 to 239: This text confused me a bit. First, the authors really have not constrained the PETM in the previous text. Second, the CIE thickness (if complete) does not allow for determination of the change in sedimentation rates. I think the authors are trying to briefly explain what John et al. (2008) determined, but as written that is unclear. I think this opening paragraph would be stronger if it clearly explained what has been determined previously in this region and by whom, and then ended with a clear list of how the subsequent discussion sections are going to elaborate on that understanding.*

We revised the paragraph and for clarity, and highlighted the PETM interval in figure 2, 3. The crucial point of this paragraph is to highlight how regionally sedimentation rate reflect hydrological cycle change in the coastal environment. For the mountainous coastal environment, higher sedimentation rates suggest more sediment supply through river runoff and precipitation shift. line 237 to 239, here we simply need to point out the limited constrains of the age model in computing changes in sedimentation rates in detail. It is worth mentioning that the thick CIE suggests relatively higher sed rates. Of course, other factors might influence accumulation rates, for example, local sea level change.

Discussion: A general observation of the discussion is that it treats the data in rather isolated silos (with the exception of using models to interpret leaf wax d2H values). And both sections 4.1 and 4.2 seem less like discussions than extensions of the associated Results. Since the Results have already been presented, then draw from all of them to drive the discussion forward. For example, I would not have a discussion section on clay mineralogy but rather one on extreme rainfall events that draws on the mineralogy and the models. That is just a suggestion, of course, and I am one to give authors latitude in how they want to tell their story! But I think a more integrated approach would ensure that the greatest added value emerges from the multi-proxy study.

> While this was intended to keep the discussion "organized" and easier to follow, we have revised the section following your suggestions, starting with a discussion of the model simulation, followed by the comparison with observations. We believe it now flows more smoothly.

*Lines 255-257: I don't think these comments quite capture the debate about clay mineralogical change at the PETM. The increase in kaolinite has been attributed to both increased humidity and more deeply erosive events; given the context of the paper, I would make those two interpretations explicit. And then… is there any evidence to distinguish between those? The model simulations (or at least what is included) suggests that extreme events and erosion are more likely explanations than increased humidity. If so, say that. Also, I'd encourage the authors to discuss the post-PETM data and allow that to inform their interpretation.*

> We clarify the clay assemblage interpretation within the context of model simulations including a mention of the post-PETM clay assemblage evidence of an enhanced hydrological cycle during the early Eocene.

*Lines 301 to 303: See comments above – the description of these records needs to recognize the analytical error and be presented with a wee bit better stratigraphic rigour.*

> Done.

*Lines 303 to 304: Ascribing the shift in d2H prior to the PETM to orbital variability seems bold. What is the evidence for this? And why don't we see similar orbital variability during the PETM? Or afterwards?*

> Background variability prior to the PETM caused by orbital forcing (~precession) has been suggested for several sections, in part supported by theory (citation: Kiel et al 2018; Lunt et al 2007; Rush et al 2021). There are several sites (e.g. Forada, Tanzania, New Zealand, Venezuela etc.) showing a positive $\delta^2H$ shift prior to PETM (line 296-297) opposite of Lodo, as well differences in the timing of weathering changes (e.g., kaolinite

increases). As each site has limited age model control, we don't really know the relative timing between locations or relative to orbital forcing. Nevertheless, as models show the regional responses to orbital forcing will differ from region to region in terms of sign of change as suggested in Campbell et al.,2023. Similar pattern also observed in other hyperthermal event (e.g. EECO) by Walters et al., 2023 based on leaf wax $\delta^2H_{n\text{-alkane}}$ spectral analysis and iCESM model output suggesting that changes in orbit forcing drive large seasonal variations in precipitation and further enhance the hydrological cycle.

*Paragraph starting line 305: This would be easier to follow if the authors clearly set out what behaviour they are attempting to interpret. I assume (but am not sure) that they are arguing that leaf wax d2H – and by extension local meteoric water, given the caveats they correctly note – does not change much in their record (barring a few anomalies). State that clearly. It will make the rest of the text easier to write and to follow. For example, it will allow the reader to understand why we are discussing different factors that could 'offset'.*

Statement modified**.**

*In addition, I feel like the authors have said that 'we have some data and there are a lot of explanations for it' without drawing on other data to try to narrow down and distinguish hypotheses. What does the mineralogy say about changes in precipitation? What do n-alkane CPIs say about reworked OM? What do ACLs (in SI but never mentioned) say about changes in vegetation? A stronger structure and a more comprehensive discussion will allow more compelling interpretations.*

This is a fair criticism. We address this with a few modifications. First, addressing shifts in plant type, we cite a lack of change in ACL and add a reference to Korasidis et al. (2022) on the regional biomes of the late Paleocene and early Eocene. Regarding reworking, we cite the CPI as well as the C isotopes which suggest minimal recorking of *n*-alkanes as opposed to what has been observed in some sections on the east coast (see Lyons et al. 2018).

*Lines 338-339: This is a really nice application of the model. But the data are not convincing. I am not convinced that there is an analytically significant shift across the PETM (see line 140). And I certainly don't think it is significant in the context of the entire record. But there is such a compelling story here! Based on other mid- and low-latitude sites, we expect a strong positive d2H shift. Assuming plants record annual precip d2H, then the authors' models also predict that. The fact that this is not seen can be resolved by considering a change in seasonal precipitation d2H and growth. That approach predicts a leaf wax d2H shift that is very small and likely below analytical error, and that is what is observed. That is a really nice finding.*

*(In fact, it is so nice that I'd like to see the authors validate it a bit – perhaps in the SI by determining if the models can predict leaf wax d2H changes at other sites. If the same approach that yields a minor shift in California also yields a minor shift in Europe and a strong positive shift in Tanzania, then that is very compelling. There has been a big opportunity missed by not using the model to assess global d2H records. Maybe for a future paper…)*

> Good points. We added three other sites (Forada, Arctic, Tanzania) to this analysis using the model predicted leaf water $\Delta\delta^2H$ change with the proxy leaf wax changes from pre-PETM to PETM, the results of which are included in the SI. As the reviewer mentioned, further validation of the proxy model is beyond the scope of this paper. We may consider a future paper for including more sites globally to discuss how seasonality and growth duration affect the leaf wax record.

*Lines 350: I like this inclusion of the d13C record.*

*End Discussion and Conclusion: The authors have a nice integrated dataset. But they never quite draw it all together into a holistic picture. For example, the d13C record is used to infer lower humidity, but that is not mentioned in the abstract or conclusions. The conclusions mention lower winter precip and slightly higher summer precip but do not make it clear that the overall annual precipitation is much lower in the 6x CO2 simulation. Picking through all of the data, it seems that there is evidence for decreased overall precipitation, especially in the winter; that the precip also becomes more episodic; that these factors and higher temperatures have combined to yield a more arid climate and that impacted the vegetation as expressed in d13C values. All of these will have contributed to a more erosive sedimentary regime. These interpretations are validated by leaf wax d2H values – but that could only be deduced with careful data-model comparison that allowed the competing controls on plant d2H to be constrained. This is a really interesting suite of data, but it does not quite come together as it could.*

> We made a number of structural changes to the discussion section and summary which should hopefully more clearly pull together the observations in the context of theory (simulations). The overall flow is improved.

*The paper has a fairly large number of grammatical errors that should be cleaned up on editing. I caught several in the abstract, but they generally become more common further into the manuscript. There are many of them in some figure captions.*

> Yes it does. We have corrected all grammatical errors.